# Video-based Human-Object Interaction Detection from Tubelet Tokens

**Danyang Tu**[1], **Wei Sun**[1], **Xiongkuo Min**[1], **Guangtao Zhai**[1(✉)], **Wei Shen**[2(✉)]

[1]Institute of Image Communication and Network Engineering, Shanghai Jiao Tong University
[2]MoE Key Lab of Artificial Intelligence, AI Institute, Shanghai Jiao Tong University
{danyangtu, sunguwei, minxiongkuo, zhaiguangtao, wei.shen}@sjtu.edu.cn

## Abstract

We present a novel vision Transforme**r**, named TUTOR, which is able to learn **tu**belet **to**kens, served as highly-abstracted spatiotemporal representations, for video-based human-object interaction (V-HOI) detection. The tubelet tokens structurize videos by agglomerating and linking semantically-related patch tokens along spatial and temporal domains, which enjoy two benefits: 1) Compactness: each tubelet token is learned by a selective attention mechanism to reduce redundant spatial dependencies from others; 2) Expressiveness: each tubelet token is enabled to align with a semantic instance, *i.e.*, an object or a human, across frames, thanks to agglomeration and linking. The effectiveness and efficiency of TUTOR are verified by extensive experiments. Results show our method outperforms existing works by large margins, with a relative mAP gain of $16.14\%$ on VidHOI and a 2 points gain on CAD-120 as well as a $4\times$ speedup.

## 1 Introduction

Human-object interaction (HOI) detection is a detailed scene understanding task, which requires both localization of interacted human-object pairs and recognition of interaction labels. Existing methods mostly investigated detecting HOIs in static images without capturing temporal information (Figure 1 (a)), thus lack the ability to detect time-related interactions (*e.g.*, `shoot` or `pass` a basketball). However, interactions are more of time-related in practical scenario, leading to a strong demand to detect HOIs from videos, a more challenging problem built on spatiotemporal semantic representations.

Transformer, originated from natural language processing (NLP), is an intuitive choice for its eminent capability of reasoning long-range dependencies, in which one of the most crucial components is the token. A token serves as an element of data representations, which is usually a word in language. However, unlike language that naturally has such a discrete signal space for building tokenized dictionaries, images lie in a continuous and high-dimensional space. To address this issue, vision Transformer (ViT) [7] provided a solution that divides each image into several local patch tokens ("visual words") to structurize the entire image as a "visual sentence"(Figure 1(b)). This solution has become a *de facto* tokenization standard followed by most existing Transformer-based methods, which has achieved excellent performance for various vision tasks, especially image classification.

Nevertheless, this patch based tokenization strategy might not be proper for video-based HOI (V-HOI) detection (as the performance degradation of the ViT-like framework shown in Table. 1a). We find that the reason is the patch tokens generated by regular splitting are difficult to exactly capture instance-level semantics (an instance is an object or a human, *e.g.*, the basketball shooter in Figure 1), which yet is crucial for V-HOI detection to reason the interaction labels. These patch tokens inevitably

---

✉Corresponding Author.

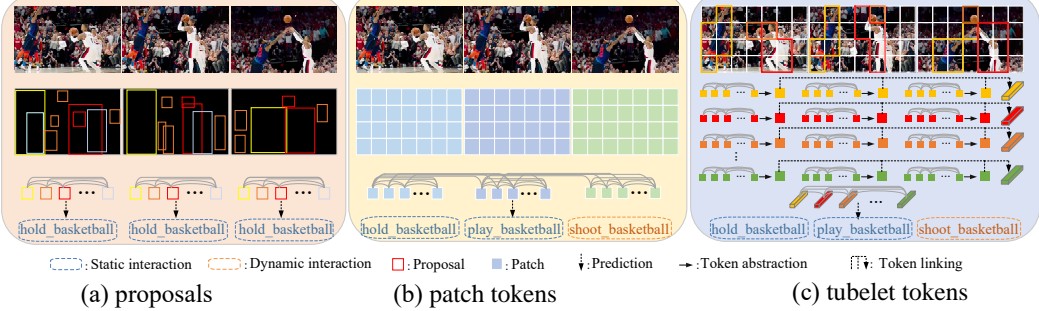

| | | |
|---|---|---|
| (a) proposals | (b) patch tokens | (c) tubelet tokens |

Figure 1: Illustration of different strategies for V-HOI detection, which are built on different representations. (a) Image-based HOI detection methods, which process each frame as *i.i.d* data, and recognize the interrelations among pre-detected proposals in each frame independently. (b) ViT-like V-HOI detection frameworks, which perform global attention mechanism on patch tokens over space and time. (c) Our proposed TUTOR, which structurizes a video into a few tubelet tokens by token abstraction along spatial domain and token linking along temporal domain.

suffer from redundancy due to an information mixture from different instances as well as insufficiency due to only a part occupancy of an instance, which limit their representation ability.

In this paper, we present TUTOR, a new Transforme**R** for V-HOI detection built on **TU**belet **TO**kens, handling aforementioned limitations favorably. The tokenization of the tubelet tokens is not based on fixed regular splitting but is jointly performed with the learning of the Transformer encoder. This enables the tubelet tokens to progressively emerge and represent high-level visual semantics. Concretely, first, along the spatial domain, we alternatively update the representation for each patch token by a selective attention mechanism and agglomerate semantically-related patch tokens into instance tokens. The selective attention mechanism ensures that attention is performed among tokens expected to belong to the same instance, which reduces redundant spatial dependencies from others. Then, along the temporal domain, we link instance tokens across frames to form the tubelet tokens. Figure 1(c) illustrates the process of tubelet token generation. Experimental results show that TUTOR outperforms existing sota methods by large margins. Specifically, we achieve a relative mAP gain of 16.4% on VidHOI [5] and a 2 points F1 score gain on CAD-120 [22], with a 4× inference speedup.

## 2 Related Work

**HOI detection.** Most previous works are devoted to detecting HOIs in static images [3, 10, 11, 13, 14, 16, 19, 21, 23, 24, 27, 29, 35, 39, 41, 42, 44, 46, 47, 50, 51, 52, 37, 54, 20, 38, 4, 48]. Without considering temporal information, these methods fail to detect time-related interactions, restricting their value in practical applications. In contrast, video-based HOI detection is a more practical problem, which however is less explored [35, 33, 34, 36, 5, 43, 18]. [35, 36, 43] detected HOIs in videos by building graph neural networks to capture spatiotemporal information. In [33], HOI "hotspots" can be directly learned from videos by jointly training a video-based action recognition network as well as an anticipation model. Inspired by image-based methods, [5] introduced a two-stage framework where the frame-wise human/object features are firstly extracted by using trajectories, and then HOIs are detected by processing the instance features as well as auxiliary features, including spatial configurations and human poses. However, these methods lack the ability to model long range contextual information, resulting in poor performance when the interacted human and object are far apart. [18] proposed a spatiotemporal Transformer to reason human-object relationships in videos, which detects human/object proposals firstly and then captures spatial and temporal information by using two dense-connected Transformers, respectively. However, such dense-connected manner introduces extra computation and ambiguity in token representation.

**Transformer in video analysis.** Transformer [40] has shown a great potential in video analysis, *e.g.*, action recognition [28, 49], video restoration [25], video question answering [12], video instance segmentation [45] and etc. However, most spatiotemporal Transformer follow the *de facto* scheme of ViT [7], *i.e.*, simply dividing an image into local patches and stacking global attention, which lacks sufficient exploration of the properties of visual signal, thus suffering from insufficiency token representation and explosive computation.

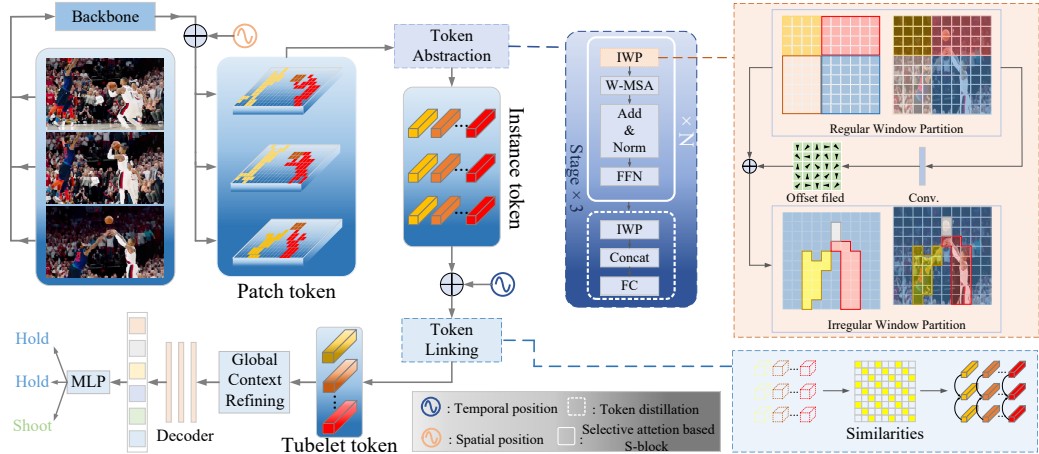

Figure 2: **The architecture of TUTOR.** It consists of 1) a backbone to generate the initial patch tokens; 2) a token abstraction module that alternatively update token representations and agglomerate patch tokens, to progressively form instance tokens; 3) a token linking module that links the semantically related instance tokens across different frames to form tubelet tokens; 4) a simple global attention layer to reinforce the global contextual information and 5) a standard Transformer decoder to decode the HOI instances. We use different dashed squares to zoom in on different key modules.

# 3 Methodology

The main idea of TUTOR is to structurize a video into a few tubelet tokens, which serve as highly-abstracted spatiotemporal representations. To this end, we propose a reinforced tokenization strategy, which jointly performs tokenization and optimization of the Transformer encoder, as illustrated in Figure 2. The process of tubelet token generation consists of two steps: 1) Token abstraction along the spatial domain, where patch tokens are alternatively updated by a selective attention mechanism and agglomerated into instance tokens; 2) Token linking along the temporal domain, where instance tokens across frames are linked to form tubelet tokens. We describe these two steps in details below.

## 3.1 Backbone

Taking a video clip $\mathbf{x} \in \mathbb{R}^{T \times H \times W \times 3}$ that consists of $T$ frames with size $H \times W$ as the input, we use a ResNet [15] followed by a feature pyramid network (FPN) [26] as the backbone on $t$-th frame to generate a feature map $\mathbf{z}_{(t,b)} \in \mathbb{R}^{\frac{H}{4} \times \frac{W}{4} \times C_0}$, where $t = 1, 2, .., T$ and $C_0 = 32$ is the channel number of the initial feature map.

## 3.2 Token Abstraction

Token abstraction is organized in 3 *Stage*s through a hierarchy of Transformer layers. Each stage performs token representation learning by a selective attention mechanism and merges semantically-related patch tokens into instance tokens by an agglomeration layer. Here, we denote the feature map of $t$-th frame inputted into $s$-th stage as $\mathbf{z}_{(t,s)} \in \mathbb{R}^{H_s \times W_s \times C_s}$. Specifically, $\mathbf{z}_{(t,1)} = \mathbf{z}_{(t,b)}$.

**Selective attention.** To eliminate the redundancy caused by information mixture from different instances, we are motivated to selectively calculate attention weights among related tokens, *i.e.*, tokens belong to the same instances. To this end, we propose an irregular window partition (IWP) mechanism (orange rectangle in the right of Figure 2), a simple yet effective strategy that samples a group of related tokens into a local window. IWP is inspired from regular window partition [30], where the tokens are grouped by sliding a regular rectangle $\mathcal{R}$ with size of $S_w \times S_w$ over the feature map $\mathbf{z}_{(t,s)}$ in $s$-th stage. For instance, $\mathcal{R} = \{[0,0], [0,1], ..., [3,4], [4,4]\}$ defines a regular window with size of $5 \times 5$. Then, for the $i$-th regular window, we have

$$\mathcal{Z}_{\text{rw}}^i = \{\mathbf{z}_{(t,s)}(\mathbf{p}_n + [x_w^i, y_w^i]) \mid \mathbf{p}_n \in \mathcal{R}\}, \tag{1}$$

where $\mathbf{z}_{(t,s)}([x,y]) \in \mathbb{R}^{1 \times C_s}$ denotes the feature vector at spatial location $[x,y]$ and $[x_w^i, y_w^i]$ is the location of the top-left point of $i$-th window. However, as shown in Figure 2, regular windows

can easily divide an instance into several parts due to the limitation of a fixed shape, leads to unrelated tokens within a window, *i.e.*, belonging to different instances. Inspired by deformable DETR [53] and deformalbe convolution [6], (the detailed comparison is described in Appendix) IWP makes a simple change by augmenting regular grid $\mathcal{R}$ with learned offsets, which allows the generated irregular windows to be aligned with humans/objects with arbitrary shapes. With offsets $\{\Delta \mathbf{p}_n | n = 1, 2, ..., N\}$ and $N = |\mathcal{R}|$, for the tokens in $i$-th irregular window, we have

$$z_{\text{irw}}^i = \{\mathbf{z}_{(t,s)}(\mathbf{p}_n + [x_w^i, y_w^i] + \Delta \mathbf{p}_n) \mid \mathbf{p}_n \in \mathcal{R}\}. \tag{2}$$

Specifically, $\Delta \mathbf{p}_n$ are learned by performing a convolutional layer with kernel size of $3 \times 3$ over the input feature map $\mathbf{z}_{(t,s)}$. As the offsets are typically fractional, the right part of Eq. 2 is implemented practically via bilinear interpolation as

$$\mathbf{z}_{(t,s)}(\mathbf{p}) = \sum_{\mathbf{q}} \mathrm{B}(\mathbf{q}, \mathbf{p}) \cdot \mathbf{z}_{(t,s)}(\mathbf{q}), \tag{3}$$

where $\mathbf{p} = \mathbf{p}_n + [x_w^i, y_w^i] + \Delta \mathbf{p}_n$ denotes an arbitrary location, $\mathbf{q}$ enumerates all neighboring integral locations, and B is the bilinear interpolation kernel.

On this basis, we alternatively update the token representation by stacking several *S-blocks* (rectangle in white solid line in Figure 2) that are built on selective attention, *i.e.*, performing attention mechanism within irregular windows. Specifically, each block is computed as

$$\hat{\mathbf{z}}_{\text{irw}}^l = \text{IWP}(\mathbf{z}^{l-1}, S_w), \tag{4}$$

$$\hat{\mathbf{z}}^l = \text{W-MSA}(\text{LN}(\hat{\mathbf{z}}_{\text{irw}}^l + \mathbf{p}_e^l)) + \text{Flatten}(\mathbf{z}^{l-1}), \tag{5}$$

$$\mathbf{z}^l = \text{Reshape}(\text{MLP}(\text{LN}(\hat{\mathbf{z}}^l)) + \hat{\mathbf{z}}^l), \tag{6}$$

where $\mathbf{z}^l$ is updated representation for all tokens, $\mathbf{p}_e^l$ is sine-based spatial position encoding at $l$-th *S-block*, and $\hat{\mathbf{z}}$ denotes various intermediate features. Here, we factorize the conventional 3D position encoding into a 2D spatial position encoding and a 1D temporal one since the spatial and temporal information are separately extracted. In detail, "W-MSA" denotes window-based multi-head self-attention, "LN" is layer normalization and "MLP" refers to multi-layer perceptron. Since the convolution layer in IWP is operated on 2D feature map yet attention is calculated on sequential features, we use a "Flatten" $(2D \rightarrow 1D)$ operation to collapse the spatial dimension and a "Reshape" $(1D \rightarrow 2D)$ operation to restore it. The computational complexity of a global MSA (G-MSA) block and an irregular-window-based block (IW-MSA) for total $T$ frames at $s$-th stage are respectively:

$$\Omega(\text{G-MSA}) = 4H_s W_s T C_s^2 + 2(H_s W_s T)^2 C_s, \tag{7}$$

$$\Omega(\text{IW-MSA}) = 4H_s W_s T C_s^2 + 2(S_w^2 + K^2) H_s W_s T C_s, \tag{8}$$

where $K = 3$ is the kernel size of convolutional layer and $S_w$ is fixed as 7. In comparison, IW-MSA can effectively reduce the computational complexity. In our experiment, the number of *S-block* for (1-3)-th stage is set to 1, 1, 3, respectively.

**Token agglomeration.** We perform token agglomeration at the end of each stage to merge semantically similar tokens. Specifically, we first perform IWP with a window size of $2 \times 2$ to dynamically sample every 4 related tokens into a window. Then, we concatenate the tokens within each window and apply a fully-connected (FC) layer on the concatenated $4C_s$-dimensional features. We set the output dimension to $2C_s$. It reduces the number of tokens by a multiple of $2 \times 2 = 4$ after each stage.

To sum up, token abstraction totally reduces the number of tokens by a factor of $4^3 = 64$ and increases the dimension by a factor of $2^3 = 8$. It structures each frame into a few instance tokens on the basis of selective attention and token agglomeration, which reduces the visual redundancy and also enjoys the advantage of Transformer with an affordable computational costs.

### 3.3 Token Linking

Assuming that after token abstraction, each frame is structured as $J$ instance tokens. Then a video clip of $T$ frames can be denoted as $z_{\text{ins}} = \{\mathbf{z}_t^j | j = 1, 2, ..., J; t = 1, 2, ..., T\}$, where $\mathbf{z}_t^j$ refers to the $j$-th token in the $t$-th frame. Here, a sine-based 1D temporal position encoding is additionally added to $z_{\text{ins}}$. The goal of token linking is to link $T$ instance tokens with the same semantic across $T$ frames, so that a video clip can be structurized as $J$ spatiotemporal tubelet tokens. To this end, we propose an exemplar

based between-frame one-to-one matching strategy. We first chose $\mathcal{Z}_q = \{\mathbf{z}_r^{j_q} | j_q = 1, 2, ..., J\}$ as the exemplar frame, where $r = \lfloor \frac{T}{2} \rfloor$ is the index of the middle frame in the video clip. We denote the tokens in the exemplar frame and those in rest frames $\mathcal{Z}_k = \{\mathbf{z}_t^{j_k} | \mathbf{z}_t^{j_k} \in \mathcal{Z}_{\text{ins}}, t \neq r\}$ as *query* tokens and *key* tokens, respectively. Then, we compute a similarity matrix $\mathbf{A}$ between the query tokens and the key tokens via a `Gumbel-Softmax` [17] operation computed over the query tokens as

$$\mathbf{A}_t^{(j_q, j_k)} = \frac{\exp(W_q^{j_q} \mathbf{z}_r^{j_q} \cdot W_k^{j_k} \mathbf{z}_t^{j_k} + \gamma_q)}{\sum_{j=1}^{J} \exp(W_q^{j} \mathbf{z}_r^{j} \cdot W_k^{j_k} \mathbf{z}_t^{j_k} + \gamma_j)} \quad s.t. \ \ t \neq r, \tag{9}$$

where $W_q^{j_q}$ and $W_k^{j_k}$ are the weights of the learned linear projections for the $j_q$-th query tokens and the $j_k$-th key tokens, respectively, and $\gamma$s are *i.i.d* random samples drawn from the `Gumbel(0,1)` distribution that enables the `Gumbel-Softmax` distribution to be close with the real categorical distribution. Then, we introduce a modified `nms-one-hot` operation to determine the one-to-one correspondence between the query tokens and the key tokens of each frame. Specifically, for $j_k$-th key token $\mathbf{z}_t^{j_k}$ in $t$-th frame, the norm `one-hot` assignment is performed by taking the value of $\texttt{argmax}\{\mathbf{A}_t^{(j_q, j_k)} | j_q = 1, 2, ..., J\}$. However, such an operation cannot ensure a one-to-one correspondence, *i.e.*, more than one key tokens in the same frame could be assigned to the same query token. To address this issue, in `nms-one-hot` scheme, for example, when $m_1$-th and $m_2$-th key token in $t$-th frame are assigned to $j_q$-th query token simultaneously, we assign the one with a higher similarity, *i.e.*, $\texttt{max}(\mathbf{A}_t^{(j_q, m_1)}, \mathbf{A}_t^{(j_q, m_2)})$, to the $j_q$-th query token. Then, if $\mathbf{z}_t^{m_1}$ has been assigned to the $j_q$-th query token, we manually set $\mathbf{A}_t^{(j_q, m_2)}$ as 0 and continue to conduct the assignment operation. Since the `nms-one-hot` operation is not differentiable, we adopt the straight through strategy in [8] to compute the assignment matrix:

$$\hat{\mathbf{A}} = \texttt{nms-one-hot}(\mathbf{A}) + \mathbf{A} - \texttt{sg}(\mathbf{A}), \tag{10}$$

where $\texttt{sg}(\cdot)$ is the stop gradient operator. $\hat{\mathbf{A}}$ is numerically equal to `nms-one-hot` assignments and its gradient is equal to the gradient of $\mathbf{A}$, which makes the token linking module differentiable and end-to-end trainable. Finally, we link the tokens corresponding to the same query token to form the tubelet tokens $\mathcal{Z}_{\text{tube}} = \{\mathbf{z}_{\text{tube}}^{j} | j = 1, 2, ..., J\}$, which is computed as

$$\mathbf{z}_{\text{tube}}^{j} = \mathbf{z}_r^{j} + W_o \frac{\sum_{t=1}^{T} \hat{\mathbf{A}}_t^{(j, \phi(j, t))} W_v \mathbf{z}_t^{\phi(j, t)}}{\sum_{t=1}^{T} \hat{\mathbf{A}}_t^{(j, \phi(j, t))}} \quad s.t. \ \ t \neq r, \tag{11}$$

where $W_o$ and $W_v$ are the learned weights of projectors, and $\phi(j, t)$ is the index of token in $t$-th frame and being assigned to $j$-th query token.

### 3.4 Global Context Refining

After token agglomeration and linking, a spatiotemporal video representation is structurized as a few tubelet tokens. On this basis, we perform an additional global attention layer to model global contextual information. Our intuition is two-fold: 1) Global context can significantly boost the performance of interaction recognition, *e.g.*, if grassland is detected, a person is more likely to be playing soccer than basketball. 2) Different interactions can be co-occurring, *e.g.*, a person is `holding` a fork could be `eating` something.

### 3.5 Decoder & Prediction Head

**Decoder.** Following the standard architecture in [2], the decoder transforms $N_q$ embeddings by stacking 6 layers consisting of self-attention and cross-attention mechanisms. These embeddings are learned position encodings which are initialized to constants and we refer them to as HOI *queries*. Being added to the input of each attention layer, the $N_q$ queries are transformed as output embeddings by the decoder, which performs global reasoning by using the entire video clip as context.

**Prediction head.** Following [37], the prediction head is composed of four feed-forward networks (FFNs): human-bounding-box FFNs $f_h$, object-bounding-box FFNs $f_o$, object-class FFNs $f_o^c$, and action-class FFNs $f_a^c$. Specifically, $f_h$ and $f_o$ are both a 3-layer perceptron followed by a sigmoid function, which output normalized human- and object-bounding box $\hat{\mathbf{b}}_h \in [0, 1]^4$, $\hat{\mathbf{b}}_o \in [0, 1]^4$,

respectively. $f_o^c$ is a linear layer followed by a softmax function, predicting the probability of object classes $\hat{\mathbf{c}}_o \in [0, 1]^{N_{obj}+1}$, where $N_{obj}$ is the number of object classes and the $(N_{obj}+1)$-th element in $\hat{\mathbf{c}}_o$ indicates the query has no corresponding human-object pair. Since actions could be co-occurring, $f_a^c$ is a linear layer followed by a sigmoid function rather than the softmax function. It outputs the probability of action classes $\hat{\mathbf{c}}_a \in [0, 1]^{N_{act}}$, which has no an additional element to indicate *no-action*. Here, $N_{act}$ is the number of action classes.

## 3.6  Loss Function

We follow the loss calculation scheme in [37], including bipartite matching and loss calculation. We describe the detailed calculating process in Appendix.

# 4  Experiments

## 4.1  Datasets & Metrics

We conduct experiments on VidHOI [5] and CAD-120 [22] benchmarks to evaluate the proposed methods by following the standard scheme. VidHOI is a large-scale dataset for V-HOI detection, comprising 6,366 videos for training and 756 videos for validation. In VidHOI, 50 relation categories are annotated, of which half are time-related ones. Mean AP (mAP) is calculated as the evaluation metric for VidHOI, which is reported over three sets: 1) Full: all 557 categories are evaluated; 2) Rare:315 categories with less than 25 instances and 3) Non-rare: 242 categories with more than 25 instances. CAD-120 is a relatively smaller dataset that consists of 120 RGB-D videos. Here, we only use the RGB images and the 2D bounding boxes annotations of humans and objects. Following standard scheme, we calculate the sub-activity F1 score as metrics.

## 4.2  Implementation Details

The dimension of HOI *query* is set to 256, which is the same as the tubelet tokens ($32 \times 2^3$). The number of queries is set to 100 for VidHOI and 50 for CAD-120. To save computational resources, the backbone is initialized by the backbone weights of QPIC [37], and then frozen without being updated. We employed an AdamW [31] optimizer for 150 epochs. A batch size of 16 on 8 RTX-2080Ti GPUs, and learning rate $l_r = 2.5e^{-4}$ for Transformer and $1e^{-5}$ for FPN are used. The $l_r$ decayed by half at 50-th, 90-th and 120-th epoch, respectively. We use a $l_r = 10^{-6}$ to warm up the training for the first 5 epochs, and then go back to $2.5e^{-4}$ and continue training.

## 4.3  Analysis of CNN-based & Transformer-based Methods

We compare CNN-based and Transformer-based methods in terms of: 1) long-range dependency modeling, 2) robustness to time discontinuity and 3) contextual relation reasoning.

**Long-range dependency modeling.** We split HOI instances into bins of size 0.1 according to the normalized spatial distances, and report the APs of each bin. As shown in Figure 3a, our Transformer-based method outperforms existing CNN-based methods in all cases, which becomes increasingly evident as the spatial distance grows. It indicates that Transformer has better long-range dependency modeling capability compared to CNN-based methods that relay on limited receptive field. With this ability, Transformer can dynamically aggregate important information from global context.

**Robustness to time discontinuity.** we randomly sample one frame every $t$ seconds from the original video to generate a new video as inputs, and report the relative performance compared to the baseline (sampling 1 frame per second). As shown in Figure 3b, the performance of CNN-based methods drop dramatically in contrast to Transformer. The main reason is that the ROI features from different frames are likely to be inconsistent due to the discontinuity of temporal domain. For Transformer, it can be partly solved by learning a variable attention weights to selectively process different features of different frames.

**Contextual relations reasoning.** we randomly pick 5 static interaction types (represented in blue) and 5 dynamic ones (green), each with over 10,000 images. Then, we calculate the average weights of self-attention in the last decoder layer on all pictures where two interactions are co-predicted. As

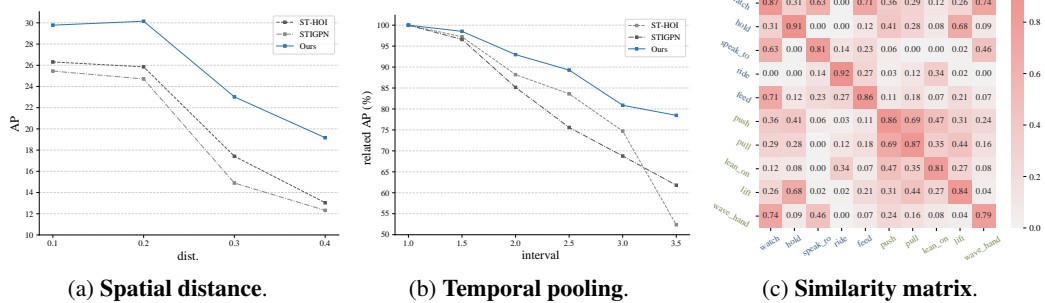

(a) **Spatial distance**.        (b) **Temporal pooling**.        (c) **Similarity matrix**.

Figure 3: The performance of CNN-based and Transformer-based methods under different scenarios.

shown in Figure 3c, Transformer can mine the interrelations among different HOI instances, *e.g.*, watch and feed, which two are likely to co-occur, get a relatively high attention weights (0.71).

### 4.4 Analysis of Token abstraction and Linking

**Token abstraction.** Table. 1a shows the influence of token abstraction, which is proposed to capture instance level representation. In comparison, CNN-based methods process cropped proposal features (Figure 1(a)), which suffers from temporal inconsistencies and the lack of contextual information, leading to the worst performance. In terms of Transformer, performance is significantly improved ( over $25\%$ ), but varies under different strategies. Interestingly, adding a simple token fusion module to ViT-like framework (ViT-like[†]), *i.e.*, fusing every 4 neighboring patch tokens after each Transformer layer, can achieve a $4\%$ relative mAP improvement. It implies that visual redundancy is an obstacle for Transformer to achieve better performance. Moreover, our irregular-window-based (IR-win) token abstraction mechanism achieves the optimal performance. Nevertheless, when replacing all irregular windows in TUTOR with regular windows (R-win), the performance is unexpectedly surpassed by ViT-like[†]. It indicates that regular windows can reduce the computational complexity, but cannot eliminate visual redundancy thoroughly.

**Token linking.** Table. 1b shows the influence of token linking. Here, the inputs for all methods are identical, which are the instance tokens generated by token abstraction module. Although computing global attentions along temporal domain without token linking achieves a competitive performance in the detection of time-related interactions, its performance is relatively poor for detecting static HOIs. We conjecture that the instance tokens in different frames are semantically similar, which introduces redundant information to static interaction detection. Moreover, the mAP decreases severely when directly use the value of `Gumbel-Softmax` as assignment weights, *i.e.*, replace the $\hat{\mathbf{A}}$ in Eq.11 with $\mathbf{A}$ in Eq.9. One possible reason is the redundancy arises within token representation due to the absence of zero value in $\mathbf{A}$. In contrast, one-hot assignment is sparse but cannot ensure an one-to-one assignment among frames, which can also cause ambiguity. In comparison, nms-one-hot assignment enforces every $T$ tokens (one per frame) to be linked, which minimizes the ambiguity and redundancy in token representation, thus achieving the optimal performance. We further investigate the effect of video length on these two assignment approaches. As shown in Figure 1c, one-hot assignment surpasses nms-one-hot when a video is longer than 16 seconds, which is caused by the simple way of choosing exemplar frame, *i.e.*, intuitively selecting the middle frame. When a video clip is long, the middle frame is semantically inconsistent with the frames that are temporally far away. We solve this problem by splitting a long video into uniform short clips and performing nms-one-hot assignment in each clip (nms-one-hot*) respectively, which yet introduces more computation.

### 4.5 Analysis of Effectiveness and Efficiency

**Effectiveness.** We verify TUTOR's effectiveness of capturing spatial and temporal semantic by observing the performance of detecting static HOI and dynamic HOI when using a quite simple decoder. For former, we use a 1-layer Transformer decoder on patch tokens in ViT-like method and instance tokens in TUTOR, respectively. the mAP is reported only on static HOI detection. As the Table 2a shows, instance tokens generated by token abstraction can stupendously improve the performance by $70\%$, compared with patch tokens. It demonstrates that token abstraction mechanism

| method | S | T | | method | S | T | |
|---|---|---|---|---|---|---|---|
| Proposal | 22.84 | 16.34 | | global | 30.07 | 19.58 | |
| ViT-like | 27.64 | 17.30 | | gumbel-softmax | 28.81 | 18.11 | |
| ViT-like$^\dagger$ | 28.45 | 18.64 | | one-hot | 30.64 | 19.27 | |
| R-win | 28.17 | 18.24 | | nms-one-hot | **32.21** | **21.28** | |
| IR-win | **32.21** | **21.28** | | | | | |

| (a) **Token abstraction**. | (b) **Token linking**. | (c) **Tokenization vs. video length**. |
|---|---|---|

Table 1: **Analysis on token abstraction and linking.** We report mAP on detecting dynamic temporal-related (T) and static spatial-related (S) HOI, respectively. ViT-like$^\dagger$ in (a) denotes that a regular-window-based token fusing is performed after each Transformer layer. Nms-one-hot* in (c) means to split a long video into several uniform short clips. Default settings are marked in gray.

| | case | mAP | | case | params | mAP | TFLOPs | FPS | speedup |
|---|---|---|---|---|---|---|---|---|---|
| spatial | w/ `TA` | **16.42** | | global | 243M | 23.51 | 0.81 | 0.5 | - |
| | w/o `TA` | 9.67 | | w/ `TA` | 104M | 25.63 | 0.42 | 1.2 | 2× |
| temporal | w/ `TL` | **8.28** | | w/ `TL` | 187M | 24.28 | 0.76 | 0.8 | - |
| | w/o `TL` | 2.30 | | w/ (`TA+TL`) | **82M** | **26.84** | **0.25** | **2.0** | 4× |

| (a) **Effectiveness**. | (b) **Efficiency**. |
|---|---|

Table 2: **Analysis on effectiveness and efficiency.** `TA` is short for token agglomeration and `TL` for token linking. We use clips of size $8 \times 384 \times 384$, with frames sampled at a rate of 1/32.

can significantly extract highly-abstracted instance level semantic, which can be easily captured even the decoder is simple. For dynamic HOI detection, we use a 4-layer perception with RELU in between as decoder. Next, we perform a global average pooling on tubelet tokens and instance tokens, which are then fed to the simple decoder to predict the dynamic interactions, respectively. Interestingly, tubelet tokens generated by token linking boost performance by $4\times$, showing us the importance to reduce the temporal redundancy.

**Efficiency.** Computing attention weights accounts for most of computational overhead in Transformer. Compared to the quadratic computational costs in global attention, we achieve a linear one. As shown in Table 2b, TUTOR achieves a $4\times$ speedup, greatly improving its usability in practical applications. Here, "FPS" is reported in terms of video, *i.e.*, the number of videos being processed per second.

## 4.6 Ablation Study

**Token agglomeration.** Token agglomeration is proposed to distill the token representation by selectively merging and projecting the semantically related tokens. We first intuitively try the k-mean [32], an excellent classical clustering algorithm, but obtain a unexpectedly poor performance, as shown in Table 3a. The reason is two-fold:1) it is difficult to integrate the k-mean with the main network into an end-to-end pipeline; 2) it is hard to determine the value of K. Then we replace irregular winodws in TUTOR with regular windows to merge every 4 neighboring tokens, an average pooling operation essentially, which reduces the feature redundancy to some extent. In comparison, irregular-window is more of an operation to selectively merge the semantically similar tokens to model highly abstracted features. It is worth emphasizing that gradually increase the dimension of agglomerated token is interestingly important, which achieves a gain of more than $6\%$ on mAP. We guess that the features are richer with increasing dimensions, as extensively adopted in CNNs.

**Window size.** Table 3b varies the window size. The instances in an image could have variant sizes. Therefore, a small-sized window is hard to overlap different instances while a large-sized one could cause information mixture as unrelated tokens may be included. We find 7 to be optimal.

**Small tricks.** We represent some small tricks for key module design in Table 3c. For `nms-one-hot` assignment, another commonly used strategy for merging assigned tokens is to concatenate them and then project them with a fully-connected layer. Compared with weighted-sum, it can slightly improve performance, but introduces more computation. For position encoding, we factorize the normally used 3D position encoding for spatiotemporal Transformer into a 2D spatial position encoding and a 1D temporal one, which two are added in token agglomeration and linking module, respectively. It

| method | S | T |
|---|---|---|
| k-mean | 26.71 | 17.56 |
| r-win | 28.82 | 18.95 |
| ir-win-C | 29.34 | 19.43 |
| ir-win-2C | **32.21** | **21.28** |

(a) **Token agglomeration**.

| size | S | T |
|---|---|---|
| 3 | 24.80 | 17.92 |
| 5 | 30.69 | 19.75 |
| 7 | **32.21** | **21.28** |
| 11 | 30.28 | 19.71 |

(b) **Window size**.

| component | trick | S | T |
|---|---|---|---|
| nms-one-hot | concat. | **32.85** | **21.44** |
| | w-sum | 32.21 | 21.28 |
| p-encoding | 3D | 31.62 | 20.59 |
| | (2+1)D | **32.21** | **21.28** |
| GCR | w/o | 31.41 | 20.63 |
| | w/ | **32.21** | **21.28** |

(c) **Small tricks**.

Table 3: **Ablation study**. In (a), "$n$C" is the dimension of agglomerated token. in (c), "p-encoding" is short for position encoding, "w-sum" for weighted-sum, "GCR" for global context refining.

| method | backbone | P | VidHOI | | | | | CAD-120 sub-activity(%) |
|---|---|---|---|---|---|---|---|---|
| | | | Full | Rare | NoneRare | S | T | |
| *CNN-based methods* | | | | | | | | |
| PMF [41] w/ SlowFast | SlowFast [9] | ✓ | 16.31 | 14.28 | 23.86 | 21.77 | 8.42 | - |
| GPNN [35] | ResNet-101 | | 18.47 | 16.41 | 24.50 | 26.41 | 16.06 | 88.9 |
| STIGPN [43] | ResNet-50 | | 19.39 | 18.22 | 28.13 | 26.58 | 18.46 | 91.9 |
| ST-HOI [5] | SlowFast | ✓ | 17.60 | 17.30 | 27.20 | 25.00 | 14.40 | - |
| *Transformer-based methods* | | | | | | | | |
| HOTR* [20] | ResNet-50 | | 21.14 | 19.83 | 30.75 | 28.36 | 9.81 | - |
| QPIC* [37] | ResNet-50 | | 21.40 | 20.56 | 32.90 | 28.87 | 9.74 | - |
| HOTR w/ SlowFast | SlowFast | | 22.84 | 21.15 | 32.86 | 27.12 | 13.29 | 90.7 |
| QPIC w/ SlowFast | SlowFast | | 22.92 | 21.64 | 33.43 | 28.41 | 13.47 | 91.3 |
| TimeSformer [1] w/ decoder | TimeSformer | | 23.17 | 21.79 | 34.57 | 27.84 | 18.90 | 92.5 |
| Ours | ResNet-50 | | **26.92** | **23.49** | **37.12** | **32.21** | **21.28** | **94.7** |

Table 4: **comparison with state-of-the-art.** P means human poses, * denotes image-based method.

is an experiential operation since spatial and temporal information are separately extracted. Global context refining, which refines global contextual information, can achieve almost 1 point gain.

## 4.7 Comparison with State-of-the-art

Unlike the popularity of image-based HOI detection, relatively less works investigate video-based one as a more practical yet challenging problem. Interestingly, the ability of image-based methods to detect dynamic HOI can be partly improved by replacing the original 2D backbone with a 3D one, but it weaken the ability of detecting static HOIs. With aforementioned strategies, our methods outperforms existing sota methods by a large margins. It is our belief that detecting HOI from video is more reasonable and practical since most interactions are time-related. Therefore, we hope our work will be useful for video-based human activity understanding research.

## 5 Discussion & Conclusion

**Limitation.** Our Transformer-based method suffers from the problem of overfitting when handling with small-scale datasets. In our experiments, we have to use the pertrained weights on VidHOI to initialize the model for CAD-120 (small scale), or performance would be severely degraded.

**Broader impacts.** We know some applications that illegally analyze user behavior by video monitoring. Therefore, strict ethical review is essential to avoid our model being used for such applications.

**Conclusion.** In this paper, we present TUTOR, a novel spatiotemporal Transformer for video-based HOI detection, which structurizes a video into a few tubelet tokens. To generate compact and expressive tubelet tokens, we propose a token abstraction scheme built on selective attention and token agglomeration, along with token linking strategy to link semantically-related tokens across frames. Our methods outperforms existing works by large margins. Going further, visual redundancy is one of the biggest obstacles for vision Transformer to achieve the same excellent performance as language Transformer, and we will devote more exploration on this in the future works.

**Acknowledgments.** This work was supported by NSFC (62225112, 61831015) and National Key R & D Program of China 2021YFE0206700, NSFC 62176159, Natural Science Foundation of Shanghai 21ZR1432200 and Shanghai Municipal Science and Technology Major Project 2021SHZDZX0102.

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

# A  Appendix

Optionally include extra information (complete proofs, additional experiments and plots) in the appendix. This section will often be part of the supplemental material.

