# Appendix for Video-based Human-Object Interaction Detection from Tubelet Tokens

**Danyang Tu**[1], **Wei Sun**[1], **Xiongkuo Min**[1], **Guangtao Zhai**[1(✉)], **Wei Shen**[2(✉)]

[1]Institute of Image Communication and Network Engineering, Shanghai Jiao Tong University
[2]MoE Key Lab of Artificial Intelligence, AI Institute, Shanghai Jiao Tong University
{danyangtu, sunguwei, minxiongkuo, zhaiguangtao, wei.shen}@sjtu.edu.cn

## A   Summary

This document contains the appendix for " Video-based Human-Object Interaction Detection from Tubelet Tokens". The primary goal of this document is to present more details about implementations and ablation studies. The codes can be found in the code supplementary files.

Concretely, we first report the data pre-processing sheme in section B. Next, in section C, we show the detailed calculation flow of "*S-block*" and "token linking" with pseudo-codes, respectively. Then, we represent some more additional ablation studies in section D, which consists of visualization of "irregular window" and "tubelet token". Finally, we describe the detailed calculating process of loss function in section E.

## B   Data Pre-processing

For TUTOR training, our default input size is 8 frames with spatial resolution of $384 \times 384$. The 8 frames are sampled from the raw video with a temporal rate of 1/32, and the starting frame is randomly sampled. In the spatial domain, we perform random resized cropping with a scale range of $[0.5, 1]$, horizontal flipping and color jitter.

## C   Detailed Calculation Flow

We illustrate the detailed calculation flow of *S-block* (algorithm 1) and token linking (algorithm 2) with pseudo-codes in Figure 1, where some special parameters for pytorch or numpy functions are omitted. We provide the original codes for TUTOR in the code supplementary files.

## D   Additional Ablation Studies

We visualize the irregular window and token linking in Figure 2 and Figure 3, respectively. It can be seen from Figure 2 that the envelope line of the sampling locations is aligned with an instance with various shapes. Besides, to generate a tubelet token, the token linking mechanism can link the semantically-related instance tokens effectively, as shown in Fiugre 3.

## E   Loss function.

**Bipartite matching.** The model outputs a fixed-size set of $N_q$ predictions, which is denoted as $\mathcal{O} = \{o^k\}_{k=1}^{N_q}$. We use the $\mathcal{G} = \{g^m | m = 1, 2, ..., M, \emptyset_1, \emptyset_2, ..., \emptyset_{N_q-M}\}$ to represent the padded groundtruths, where $M$ is the real number of HOI instances in a video clip and $\emptyset$ refers to *no-instance*.

36th Conference on Neural Information Processing Systems (NeurIPS 2022).

**Algorithm 1** Pseudocode of S-block in a PyTorch-like style.

IWP: irregular window partition; LN: layer normalization; WR: window reverse.

```
#z: input feature map, BxTxHxWxC
#S_w: the size of window
#z_o: output feature map

z = z.view(BT,HW,C) #B:batch size
z_l = LN(z)

#BT,HW,C -> BT*num_windows, S_w, S_w, C
#number_windows = HW / (S_w_i x S_w_i)
z_l = z_l.view(BT,H,W,C)
z_l = z_l.permute(0, 3, 1, 2).contiguous()
z_irw = IWP(z_l, s_w)

#Window-based Multi-head self-attention
z_i = W-MSA(z_irw)

#BT*num_windows, S_w, S_w, C -> BT,HW,C
z_i = WR(z_i).view(BT,HW,C)

z_o = z + z_o
z_o = z_o + FFN(LN(z_o))
z_o = z_o.view(B,T,H,W,C)
```

**Algorithm 2** Pseudocode of token linking in a PyTorch-like style.

mul: matrix multiplication; sum: summation.

```
#z: input feature map, BxTxNxC
#z_o: output feature map

# index of exemplar frame
r = floor(T)

#BxNxC
tokens_q = z[:,r,:]

#Bx(T-1)xNxC
tokens_k = delete(z, 1, r)

#Bx(T-1)xNxN
matrix_sim = gumbel-softmax(tokens_q, tokens_k)
matrix_sim = nms-one-hot(matrix_sim)

#Bx(T-1)xNxC
z_k = mul(matrix_sim,tokens_k)

z_o = tokens_q + sum(z_k, axis=1)
```

Figure 1: The pseudocode of S-block and token linking.

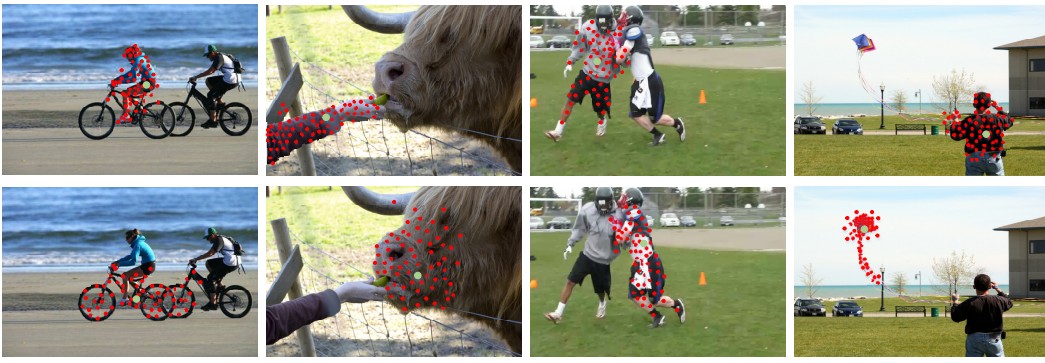

Figure 2: **Visualization of irregular windows.** Each green point is the location of an instance token obtained by token token abstraction, and the red points are the locations of the tokens involved in agglomeration. There are totally $4^3 = 64$ sampling points for each instance token, since every 4 tokens are agglomerated in a token agglomeration layer and 3 layers are employed. As the input of TUTOR are feature maps ouputted by the backbone, the locations in the original image are calculated via bilinear interpolation.

Then, the matching process can be formulated as an injective function: $\omega_{T \to O}$, being computed as

$$\mathcal{L}_{cost} = \sum_{k=1}^{N_q} \mathcal{L}_{match}(g^k, o^{\omega(k)}), \tag{1}$$

where $\omega(k)$ is the index of predicted HOI instance assigned the $i$-th groundtruth, and $\mathcal{L}_{match}(t^k, o^{\omega(k)})$ is the matching cost between the $k$-th groundtruth and $(\omega(k))$-th prediction, being calculated as

$$\mathcal{L}_{match}(g^k, o^{\omega(k)}) = \alpha_1 \mathcal{L}_b^k + \alpha_2 \mathcal{L}_o^k + \alpha_3 \mathcal{L}_a^k. \tag{2}$$

$\mathcal{L}_b$ is regression cost between groundtruths $\mathbf{b}$ and predictions $\hat{\mathbf{b}}$ for human and object boxes:

$$\mathcal{L}_b^k = \beta_1 [\|\mathbf{b}_h^k - \hat{\mathbf{b}}_h^{\omega(k)}\| + \|\mathbf{b}_o^k - \hat{\mathbf{b}}_o^{\omega(k)}\|] - \beta_2 [\text{GIoU}(\mathbf{b}_h^k, \hat{\mathbf{b}}_h^{\omega(k)}) + \text{GIoU}(\mathbf{b}_o^k, \hat{\mathbf{b}}_o^{\omega(k)})]; \tag{3}$$

$\mathcal{L}_o$ is recognition cost between groundtruths $\mathbf{c}_o$ (one-hot vector) and predictions $\hat{\mathbf{c}}_o$ for objects:

$$\mathcal{L}_o^k = -\hat{\mathbf{c}}_o(\omega(k)) \quad s.t. \quad \mathbf{c}_o(k) = 1; \tag{4}$$

$\mathcal{L}_a$ is recognition cost between groundtruths $\mathbf{c}_a$ and predictions $\hat{\mathbf{c}}_a$ for actions:

$$\mathcal{L}_a^k = -\frac{1}{2} \left( \frac{\mathbf{c}_a^{k\top} \hat{\mathbf{c}}_a^{\omega(k)}}{\|\mathbf{c}_a^k\|_1 + \epsilon} + \frac{(1 - \mathbf{c}_a^k)^{\top}(1 - \hat{\mathbf{c}}_a^{\omega(k)})}{\|1 - \mathbf{c}_a^k\|_1 + \epsilon} \right), \tag{5}$$

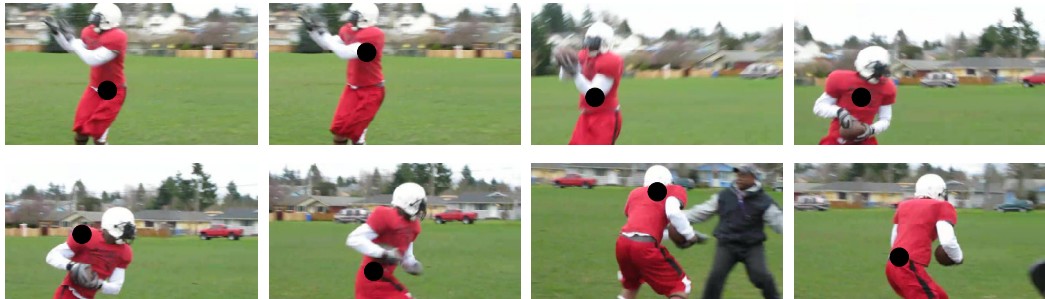

Figure 3: **Visualization of token linking.** The black points in different frames are the locations of the instance tokens involved in token linking, *i.e.*, the instance tokens being linked in the same tubelet token.

which takes both positive and negative action classes into account by using the weighted average of the two with the inverse number of nonzero elements as weight rather than the vanilla one. Finally, we follow DETR [3] to use Hungarian algorithm [1] to determine the optimal assignment $\hat{\omega}$ among the set of all possible permutations of $N_q$ elements $\mathbf{\Omega}_{N_q}$, which is formulated as

$$\hat{\omega} = \underset{\omega \in \mathbf{\Omega_{N_q}}}{\arg\min} \, \mathcal{L}_{cost}. \tag{6}$$

**Loss calculation.** After the optimal one-to-one matching between the groundtruths and the predictions is found, the loss to be minimized in the training phase is calculated as $\mathcal{L} = \eta_1 \mathcal{L}_b + \eta_2 \mathcal{L}_o + \eta_3 \mathcal{L}_a$, where $\mathcal{L}_b$ is defined as the same as Eq. 3 and $\mathcal{L}_o$ is the CE loss and $\mathcal{L}_a$ is the focal loss.

Following [2], we set $\alpha_1$, $\alpha_2$, $\alpha_3$ to 1, 1, 1, and $\beta_1$, $\beta_2$ to 2.5, 1, respectively. For loss calculation, $\eta_{(1,2,3)}$ are all set to 1.

# F  Comparison with Deformable DETR

TUTOR and deformable DETR are designed for different tasks with different objectives, which leads to a significant difference between TUTOR and deformable DETR. Specifically, deformable DETR maintains a fixed number of tokens to capture fine-grained features. Although it calculates attention locally, there is no explicit module to agglomerate a group of related tokens. In comparison, TUTOR involves a clustering process, where related tokens are grouped into an irregular window and then updated and agglomerated. These two different designs are proposed to meet different task requirements. Object detection demands location-sensitive features to precisely regress the bounding boxes while the final goal of HOI detection is to classify the interaction categories, which requires location-invariant features within the union region of a pair of interacted human and object. Consequently, deformable DETR is suitable for object detection but might not for HOI detection. Without an explicit clustering mechanism, deformable DETR cannot capture instance-level tokens, which prevents high-level understanding of interactions.

# G  Token Linking with Hungarian algorithm

Hungarian algorithm can be also used to find one-to-one matching, which has been tried in our early experiments. It achieved a slightly lower result than nms-one-hot (26.04 vs. 26.92 on Full set in VidHOI). We conjecture that the reason is as follow. For the best case, i.e., not more than one key token in each frame is assigned to the same query token, the results of nms-one-hot and the Hungarian algorithm are the same. For other cases, nms-one-hot prioritizes matching key token with higher similarity to the corresponding query token while Hungarian algorithm aims at maximizing the sum of global similarities after matching. However, in a video clip, humans and some salient objects are the protagonists of the scene while background objects are not of our interests. For example, in Figure 1, the basketball player 1 (in blue short), the player 2 (in white short) and the basketball are the protagonists while the background people have little effects to detect the HOIs (hold/play/shoot basketball). These protagonists have more distinguishable features, so they are more likely to have a higher similarity across different frames. Thus, nms-one-hot can correctly and preferentially match

these protagonists, but the Hungarian algorithm may fail to do so, and may even match the player 1 to some background humans in different frames to get the global maximum. Overall, nms-one-hot explicitly matches the protagonists first, while the Hungarian algorithm treats all objects equally.