# OpenReview forum: "Video-based Human-Object Interaction Detection from Tubelet Tokens"
_NeurIPS.cc/2022/Conference — NeurIPS 2022 Accept_

### Official Review · Reviewer_Jbmu · 2022-07-03

**Rating:** 5
**Confidence:** 3
**Soundness:** 3 good
**Presentation:** 3 good
**Contribution:** 3 good

**Summary:**

This paper proposes a new method for obtaining efficient token representations in transformer-based HOI detection in videos. The key idea is to spatially aggregate feature tokens using irregular window partition in order to deal with various shapes of objects (token abstraction) and to temporally link the tokens based on the similarity (token linking). The proposed method is evaluated on benchmark datasets and achieved favorable performance.

**Questions:**

1. I think one reasonable way to find one-to-one matching in token linking module is to use Hungarian algorithm. Is it possible to use it, and if so, how does it compare to the proposed scheme for linking tokes?
1. In the token linking module, it is implicitly assumed that a token in a frame always match to another token in another frame. However, I think this is not always the case because an object may be occluded after some frames or an object may appear at some frame. In this view, I do not think it is optimal to try to find one-to-one matching between frames. I would like to know the authors’ opinion on this point.


**Limitations:**

The limitations and potential negative impact are discussed in Section 5.

**Strengths And Weaknesses:**

## Strength
1. The proposed framework is overall reasonable and sound. Especially the design of token linking module is new. I think the proposed method is a reasonable extension of transformer-based HOI detection in videos.
1. The proposed method achieved favorable results on standard benchmark datasets.
1. The effectiveness of the proposed components are verified at least to some extent.

## Weakness
1. The proposed method is not discussed well under the context of very similar work that should have inspired the proposed method. IWP in token abstraction module is very similar to deformable DETR [52]. The appropriate credit should be given to [52] and maybe also to [A1], and the proposed method should be discussed in relation to these works. In my view, token abstraction module is not a innovation of the present paper, rather, it is adaptation of the existing concept.

1. The experiment is not very satisfactory.
    1. There is no ablation study on global context refining.
    1. Section 4.5: The reason why different decoders (1-layer, 4-layer) are used is not explained. How is the performance differences if the normal 6-layer decoder is used?
    1. Figure 3c: I find the explanation of this figure is not convincing enough. In L256 the example of watch and feed is picked up, but watch is more strongly related to wave hand, which I think is strange. Overall, I find difficulty in drawing any solid conclusion from this similarity matrix.

1. Some of the important details are not clear and confusing.
    1. In table 1a, what is the concrete process of “ViT-like” variant?
    1. In table 1b, does the “Gumbel-softmax” variant ensure one-to-one assignment?
    1. In equation 1, “[(i-1)S_w, (i-1)S_w]”, does this mean the regular windows are placed only diagonally? Isn’t it something like “[(i-1)S_w, (j-1)S_w]”?
    1. In L102, “i-th regular window” is this “i” different from “i” in L103 ([i,j], C_i)? If so, this is very confusing.
    1. L113-114, “q enumerates all integral locations”. What does “all integral locations” mean? How are they calculated?
    1. L144: Is “N” here equal to “N” in L109?
    1. L145: I recommend not using index “i” for here and there to avoid confusion.
    1. L150: Is “q” here mistake for “r”?
    1. The caption of Table 1: What is “global$^{\dagger}$”?
    1. How the accuracies of other methods in Table 4 are obtained?

1. I’m afraid the paper, not significantly but slightly, violates the formatting instruction of NeurIPS. The margin seems too small and the characters in figures are invisible without zooming in (especially Fig. 3 and Tab.1c). I’m afraid this is not fair.

1. Typos and editorial errors.
    1. L163 “the the”.
    1. L284 “frame. when…”.
    1. L292 “respectively.
    1. the mAP…”.
    1. Table 4: wrong citation for QPIC.
    1. [52] ICLR 2021, not 2020.
    1. Checklist 4b: Since existing assets are used (according to the answer to 4a), the answer to this question should be either yes or no. Actually it should be “No” because I found no license statement.

[A1] Dai+, Deformable Convolutional Networks, ICCV 2017

---

> ### Author Response · Authors · 2022-08-02
> **Response to Reviewer Jbmu [1/3]**
>
> Thanks for your valuable comments and your appreciation for our technical contributions. Below we discuss the concerns you have raised in detail.
>
> **Q1: The relation to deformable DETR.**
>
> Deformable DETR [52] indeed inspired TUTOR and we give more credit to [52] and [A1] in the revised version. However, TUTOR and deformable DETR are designed for different tasks with different objectives, which leads to a significant difference between TUTOR and deformable DETR. Specifically, deformable DETR maintains a fixed number of tokens to capture fine-grained features. Although it calculates attention locally, there is no explicit module to agglomerate a group of related tokens. In comparison, TUTOR involves a clustering process, where related tokens are grouped into an irregular window and then updated and agglomerated. These two different designs are proposed to meet different task requirements. Object detection demands location-sensitive features to precisely regress the bounding boxes while the final goal of HOI detection is to classify the interaction categories, which requires location-invariant features within the union region of a pair of interacted human and object. Consequently, deformable DETR is suitable for object detection but might not for HOI detection. Without an explicit clustering mechanism, deformable DETR cannot capture instance-level tokens, which prevents high-level understanding of interactions.
>
> In our early experiments, we have tried to replace the token abstraction module in TUTOR with deformable DETR, but the performance is relatively poor (more than 10% degradation compared to TUTOR). Overall, TUTOR is more suitable for high-level scene understanding tasks, such as HOI, with the ability of extracting instance tokens, while deformable DETR shows a better ability in extracting fine-grained features.
>
> It is worth mentioning that few Transformer-based works discussed the problem of dynamically clustering, especially in the HOI community. Therefore, we hope that TUTOR can introduce some new ideas to the community.
>
> ---
>
> **Q2: Ablation study on global context refining.**
>
> Global context refining (GCR) achieves a performance gain of 0.63 on the Full set of VidHOI (26.29 w/o GCR vs. 26.92 w/ GCR). The detailed results are added in Table 3c in the revision.
>
> ---
>
> **Q3: The reason why different decoders are used in Section 4.5.**
>
> As explained in line 289-291, we conducted this experiment to verify the effectiveness of instance tokens (token abstraction) and tubelet tokens (token linking) in capturing spatial and spatiotemporal information, respectively. Therefore, we tried to use a very simple decoder, since an over-powerful decoder would make it hard to verify whether the performance gain comes from better feature extraction or powerful decoding capabilities.
>
> More specifically, since 1-layer is the simplest architecture of a decoder, we used it to test the performance of instance tokens on static HOI detection. Besides, for time-related HOI detection, we further removed the Transformer decoder and directly used a global average pooling layer followed by a 4-layer MLP, i.e., 4 fully-connected layers (not 4-layer decoder that consists of self- and cross-attention), to detect the time-related interactions. As shown in Table 2a, as the decoder is fairly simple, the performance gain between instance/tubelet tokens and patch tokens become more evident, which verifies the effectiveness of token abstraction and token linking in capturing spatial and temporal information.

---

> > ### Author Response · Authors · 2022-08-02
> > **Response to Reviewer Jbmu [3/3]**
> >
> > **Q10: Using Hungarian algorithm to find one-to-one matching in token linking.**
> >
> > Yes, Hungarian algorithm can be used to find one-to-one matching, which has been tried in our early experiments. It achieved a slightly lower result than nms-one-hot (26.04 vs. 26.92 on Full set in VidHOI). We conjecture that the reason is as follow. For the best case, i.e., not more than one key token in each frame is assigned to the same query token, the results of nms-one-hot and the Hungarian algorithm are the same. For other cases, nms-one-hot prioritizes matching key token with higher similarity to the corresponding query token while Hungarian algorithm aims at maximizing the sum of global similarities after matching. However, in a video clip, humans and some salient objects are the protagonists of the scene while background objects are not of our interests. For example, in Figure 1, the basketball player 1 (in blue short), the player 2 (in white short) and the basketball are the protagonists while the background people have little effects to detect the HOIs (hold/play/shoot basketball). These protagonists have more distinguishable features, so they are more likely to have a higher similarity across different frames. Thus, nms-one-hot can correctly and preferentially match these protagonists, but the Hungarian algorithm may fail to do so, and may even match the player 1 to some background humans in different frames to get the global maximum. Overall, nms-one-hot explicitly matches the protagonists first, while the Hungarian algorithm treats all objects equally.
> >
> > ---
> >
> > **Q11: One-to-one matching may not be the optimal solution when an object disappear in some frames.**
> >
> > As aforementioned in **G1**, we have used two different strategies to alleviate the errors that introduced by one-to-one matching. However, the optimal solution should rely on a thorough and completely correct understanding of each frame, which we have to admit is a quite difficult task. We will keep working on this issue in the future.

---

> > > ### Comment · Reviewer_Jbmu · 2022-08-08
> > > **Many concerns addressed but some additional comments**
> > >
> > > I appreciate the authors' detailed feedback.
> > > It indeed helps me better understand the contribution of the paper.
> > >
> > > ---
> > > **Q1**
> > > > Object detection demands location-sensitive features to precisely regress the bounding boxes while the final goal of HOI detection is to classify the interaction categories, which requires location-invariant features within the union region of a pair of interacted human and object.
> > >
> > > I think this is a very good argument that helps readers better understand the different characteristics of object detection and HOI detection as well as the rationality of the design of the proposed method. I recommend adding this kind of description in the manuscript.
> > >
> > > In addition, I understood the main difference with deformable DETR lies in token agglomeration part. If that is the case, I think it is better to clarify it in the manuscript.
> > >
> > > ---
> > > **Q2**
> > >
> > > Thank you but I could not find the following in Table 3c.
> > > > (26.29 w/o GCR vs. 26.92 w/ GCR)
> > >
> > > Is there any error?
> > >
> > > ---
> > > **Q3**
> > >
> > > I understood the authors' intention, but I still have a concern on the experimental design.
> > > > Therefore, we tried to use a very simple decoder, since an over-powerful decoder would make it hard to verify whether the performance gain comes from better feature extraction or powerful decoding capabilities.
> > >
> > > I think this implies the effectiveness of the proposed components are not substantial enough that the effect diminishes if they are used with powerful decoder.
> > > Since the proposed method relies on the powerful decoder of 6-layers as mentioned in Section 3.5, I think the effectiveness of each component should be discussed in this setting.
> > >
> > > ---
> > > **Q4**
> > >
> > > I am still not sure if the co-occurrence of "watch" and "wave hand" is reasonable or not. Currently the rationality of the similarity matrix is claimed only on the basis of the subjective experience, i.e.,
> > > > they are likely to co-occur according to our experience
> > >
> > > I wonder if it is possible to show more objective evidence such as the frequency of the co-occurrence calculated by the ground-truth labels.
> > >
> > > ---
> > > **Q5-11**
> > >
> > > Thank you for the feedback, I understood them.
> > > I think the explanation in the response to Q10 and Q11 contain good insights to the readers.
> > > Therefore I recommend adding these descriptions in the manuscript or supplementary material.
> > >
> > > ---
> > > **Overall**
> > >
> > > Since many of my concerns are addressed, I increased my rating accordingly.
> > > If the authors will provide further revision, I would appreciate if the revised parts are highlighted, which helps the reviewers easily find out the revised parts.

---

> > > > ### Author Response · Authors · 2022-08-09
> > > > **Response to jbmu**
> > > >
> > > > Thank you so much for your response and appreciation.
> > > >
> > > > ---
> > > >
> > > > **Q1**
> > > >
> > > > Thank you for you appreciation. Yes, the main difference with deformable DETR lies in the token agglomeration part. The detailed discussion will be added in the Appendix, including the difference between object detection and HOI detection, as well as the difference between TUTOR and deformable DETR.
> > > >
> > > > ---
> > > >
> > > > **Q2**
> > > >
> > > > No, there is no error here. As the interaction categories in VidHOI are divided into two different types: static interactions and time-related interactions. Therefore, in Table 3c, we detailed reported the ablation results of global context refining (GCR) on two subsets of VidHOI for detecting static interactions (S) and time-related interactions (T), respectively. The ablation result in our response (i.e., 26.29 w/o GCR vs. 26.92 w/ GCR) is the overall result of GCR on the whole VidHOI.
> > > >
> > > > ---
> > > >
> > > > **Q3**
> > > >
> > > > Thank you for your valuable comments. We understand your concern. To make the results substantial enough, we further reported the results of 2-layer and 4-layer Transformer decoders (the results of 6-layer are reported in table 4). The results are shown below.
> > > >
> > > > | Decoder |   |    S         |  |    T       |
> > > > |:-------:|:-----:|:---------:|:-----:|:-------:|
> > > > |         | Patch | Instance  | Patch | Tubelet |
> > > > | MLP     | ---   | ---       | 2.30  | 8.28    |
> > > > | 1-layer | 9.67  | 16.42     | ---   | ---     |
> > > > | 2-layer | 12.35 | 18.27     | 6.24  | 14.85   |
> > > > | 4-layer | 19.40 | 25.96     | 10.32 | 18.67   |
> > > > | 6-layer | 27.12 | 32.21     | 13.29 | 21.28   |
> > > >
> > > > Here, the patch tokens are generated by HOTR [19] with SlowFast as backbone. Two conclusions can be drew from the results: 1) The decoder is important as a stronger decoder can achieve better results, both for patch and instance/tubelet tokens; 2) The performance gain between instance/tubelet tokens and patch tokens becomes more evident as the decoder becomes simpler, which verifies the effectiveness of token abstraction and token linking in capturing spatial and temporal information. These results will be added in the Appendix.
> > > >
> > > > ---
> > > >
> > > > **Q4**
> > > >
> > > > As a simple validation, we count all frames used for drawing Figure 3c and calculated the frequency of the co-occurrence by using the ground-truth labels. We reported the percentage of frames where two interaction co-occur over all frames.
> > > >
> > > > |           | watch | hold | speak-to | ride | feed | push | pull | lean-on | lift | wave-hand |
> > > > |:---------:|:-----:|:----:|:--------:|:----:|:----:|:----:|:----:|:-------:|:----:|:---------:|
> > > > | watch     | 76    | 27   | 59       | 4    | **68**   | 24   | 17   | 7       | 12   | **72**        |
> > > > | hold      | 27    | 72   | 0        | 6    | 0    | 52   | 14   | 0       | 69   | 0         |
> > > > | speak-to  | 59    | 0    | 68       | 0    | 17   | 0    | 0    | 0       | 0    | 42        |
> > > > | ride      | 4     | 6    | 0        | 81   | 0    | 0    | 0    | 0       | 0    | 0         |
> > > > | feed      | 68    | 0    | 17       | 0    | 71   | 0    | 0    | 0       | 0    | 2         |
> > > > | push      | 24    | 52   | 0        | 0    | 0    | 63   | 0    | 0       | 31   | 0         |
> > > > | pull      | 17    | 14   | 0        | 0    | 0    | 0    | 61   | 0       | 28   | 0         |
> > > > | lean-on   | 7     | 0    | 0        | 0    | 0    | 0    | 0    | 64      | 0    | 0         |
> > > > | lift      | 12    | 69   | 0        | 0    | 0    | 31   | 28   | 0       | 61   | 0         |
> > > > | wave-hand | 72    | 0    | 42       |  0    | 0    | 2    | 0    | 0       | 0    | 79       |
> > > >
> > > >  As the results shown, the attention scores in Figure 3c and the frequency of the co-occurrence are mostly consistent.
> > > >
> > > > ---
> > > >
> > > > **Revision**
> > > >
> > > > We highlighted the revised parts in the revision and all of aforementioned discussion were added in appendix.
> > > >
> > > > Thank you again for your detailed comments. Your valuable comments help us to make our paper stronger.

---

> > ### Author Response · Authors · 2022-08-02
> > **Response to Reviewer Jbmu [2/3]**
> >
> > **Q4: Question about Figure 3c, and why is "watch'' and "feed'' picked up, but not the "watch'' and "wave hand''?**
> >
> > In **G2**, we described the detailed process of generating the similarity matrix and explained the conclusion we draw from Figure 3c. To sum up, Transformer can mine the priors of these co-occurring interactions, i.e., interactions that are more likely to co-occur have higher attention score, which can improve the performance of HOI detection since it is a multi-label classification task.
> >
> > We just took "watch'' and "feed'' as the examples, since 1) they are likely to co-occur according to our experience and 2) the attention score between them is high. In a similar manner, we can also choose the "watch" and "wave hand" as the examples. The consistency between co-occurring and high attention scores evidences TUTOR can mine the priors of these co-occurring interactions.
> >
> > In addition to the above conclusion, there is another purpose for showing Figure 3c. Cross-attention has attracted lots of interests and has been widely explored while self-attention in decoder was under-explored in Transformer-based HOI detection methods. We verified that self-attention in the decoder can capture the priors of co-occurring interactions and these priors are important for HOI detection. Therefore, our experiment may help the community to explore more potential of self-attention in the decoder, e.g., some well-designed HOI queries (randomly initialized for now) that allow self-attention to learn more priors.
> >
> > ---
> >
> > **Q5: What is the concrete process of "ViT-like" variant?**
> >
> > ViT-like is illustrated as Figure 1b, i.e., splitting an image into a fixed number of patches. On this basis, an additional token fusion layer is added at the end of each ViT layer in the ViT-like variant. The fusion layer fuses every 4 neighboring patch tokens by taking their average value as the output (Line 262-263), which is similar to the average pooling in CNN and can reduce the spatial redundancy to some extent.
> >
> > ---
> >
> > **Q6: Does Gumbel-softmax variant ensure one-to-one assignment?**
> >
> > No, it does not ensure one-to-one assignment.  As explained in Line 275-278, when directly using the value of Gumbel-softmax as assignment weights (replacing the $\hat{\mathbf{A}}$ in Eq.11 with $\mathbf{A}$), all key tokens in a frame are weighted added into a query token.
> >
> > ---
> >
> > **Q7: What does "all integral locations" in Line 113-114 mean?**
> >
> > Equation 3 follows the standard process of bilinear interpolation. Specifically, the predicted offsets are usually fractional, which leads to a fractional sampling location that can not be directly sampled from regular grids on feature maps. In this case, its value is calculated by the weighted sum of its neighbouring 4 pixels with integral locations. For example, if the target location is [3.5, 4.1], the locations of neighbouring pixels would be [3,4] (top-left), [4,4] (top-right), [3,5] (bottom-left) and [4,5] (bottom-right).
> >
> > ---
> >
> > **Q8: typos and confusing details**
> >
> > Thank you. We have carefully revised our manuscript and re-uploaded a revision. Specifically, we rewrite equation 1 and use $[w_x^i, w_y^i]$ to denote the location of the top-left point of the $i$-th window. If we count windows row-by-row, $w_x^i = (i \cdot S_w) $% $W$ and $w_y^i = \lfloor\frac{i \cdot S_w}{W} \rfloor \cdot S_w$, where % is remainder operation. We replace $N$ in Line 144 with $J$, which indicates the number of instance tokens in each frame after token abstraction, and $j$ is the index of instance token. $q$ in Line 150 is revised as $r$ and $\text{global}^{\dagger}$ in the caption of table 1 is revised as $\text{ViT-like}^{\dagger}$. We have carefully double-checked all equations to avoid confusion.
> >
> > We did use ``vspace" to adjust the spacing between some paragraphs in our reviewed version, but the margin did not changed. In revision, we have removed them all. Besides, Figure 3 and Table 3c have been enlarged for clarity in the revision. In checklist 4b, VidHOI and CAD-120 are both public databases, but we did not find specially mentioned license. This is why we answered this question as N/A. We have revised it as No in revision.
> >
> > ---
> >
> > **Q9: How the accuracies of other methods in Table 4 are obtained?**
> >
> > The results of ST-HOI [4] are quoted from [4]. For other methods, we use their official codes to re-trained their models on the two datasets, vidHOI and CAD-120, except for the results of [35, 42] on CAD-120, which are quoted from [35, 42]. The training strategies and data augmentation are kept as the same as TUTOR.

---

### Official Review · Reviewer_JsuZ · 2022-07-12

**Rating:** 5
**Confidence:** 4
**Soundness:** 2 fair
**Presentation:** 3 good
**Contribution:** 2 fair

**Summary:**

This paper presents a novel vision Transformer TUTOR for human-object interaction detection in videos. TUTOR structurizes a video into a few tubelet tokens by agglomerating and linking semantically-related patch tokens along spatial and temporal domains. Experiments are conducted on VidHOI and CAD-120.

**Questions:**

See above

**Limitations:**

See above

**Strengths And Weaknesses:**

Strengths:
1. Ablation studies are conducted to evaluate the effectiveness of each component.
2. The proposed method achieves a new state-of-the-art performance.

Weaknesses:
1. In the paper, token linking aims at generating complete tubelets whose length is equal to the duration of the video. However, in real scenarios, the instances don’t necessarily appear in the entire video and the tubelets are thus broken into fragments. In these cases, generating a full-length tubelet is unreliable and may introduce much noise.
2. Some closely related works, such as [17], should also be compared in Table 5 and 6.
3. The manuscript is not well written and needs to be revised carefully.

---

> ### Author Response · Authors · 2022-08-02
> **Response to Reviewer JsuZ**
>
> Thanks for your valuable comments and your appreciation for our technical contributions. Below we discuss the concerns you have raised in detail.
>
> ---
>
> **Q1: Problem about missing instances in some frames.**
>
> We answer this question in **G1**. As aforementioned, TUTOR uses two simple but effective strategies to address this problem, including splitting a long video into several short ones, and summing the matched tokens by using similarities as weights, instead of concatenating them directly. But we agree that it is still not the optimal solution and it deserves more exploration to completely eliminate the redundancy, especially on the temporal domain.
>
> ----
>
> **Q2: [17] should also be compared in Table 5 and 6.**
>
> We show our respect to [17] and have briefly introduced it in related work (Line 63-66). However, its source code is not available. Our reproduced code does not guarantee a fair comparison, especially the speed of inference.
>
> ----
>
> **Q3: Revise the manuscript.**
>
> Thank you. We have carefully double-checked our manuscript and revised it. The revision is re-uploaded.

---

> > ### Comment · Reviewer_JsuZ · 2022-08-08
> > **Final decision**
> >
> > The response addressed some of my concerns such as token linking. But the remaining issues such as performance comparison remain unresolved. Overall, I stick to my rating of borderline accept.

---

> > > ### Author Response · Authors · 2022-08-09
> > > **Response to Reviewer jsuZ**
> > >
> > > Thank you so much for your response and appreciation. It is worth emphasizing that [17] is not specifically proposed for V-HOI detection, but for dynamic scene graph generation (DSGG), which aims to detect the object relationships in a video. Note that, DSGG consists of only several action categories but mostly spatial relationships (e.g., above, behind). In comparison, the human-object relationships are mostly described by actions in V-HOI detection. Due to the task gap, [17] did not reported the results on VidHOI and CAD-120 (for HOI detection). Besides, its source code is not available. This is why we introduced it in related work but did not compared it in table 4.
> > >
> > > Fortunately, we find another related method STTran [A2], which reported better results than [17] on DSGG. The official code of STTran [A2] is available. So we compare our method with STTran [A2] on VidHOI instead.
> > >
> > > Concretely, we fine-tuned it for 50 epoches by using the same data augmentation strategy as TUTOR. The results of STTran and TUTOR on VidHOI are shown as below.
> > >
> > > |        | Full  | Rare  | NoneRare | S     | T     |
> > > |:------:|:-----:|:-----:|:--------:|:-----:|:-----:|
> > > | Strran | 20.38 | 18.64 | 29.89    | 27.04 | 12.95 |
> > > | TUTOR  | 26.92 | 23.49 | 37.12    | 32.21 | 21.28 |
> > >
> > > The results show that the method for DSGG can be applied for V-HOI detection but achieves inferior performance. In table 4, we thoroughly compare the different methods, including the methods that specifically proposed for V-HOI detection [40, 35, 42, 4], recent image-based methods[19, 37] and their variants (using Slowfast as backbone to extract temporal information), and popular Transformer-based video analysis method [1].
> > >
> > > Overall, thank you for your valuable comments again,  we hope our response can address your concerns.
> > >
> > > ---
> > >
> > > [A2] Yuren+. Spatial-Temporal Transformer for Dynamic Scene Graph Generation. In ICCV, 2021.

---

### Official Review · Reviewer_7mrZ · 2022-07-13

**Rating:** 6
**Confidence:** 4
**Soundness:** 3 good
**Presentation:** 3 good
**Contribution:** 3 good

**Summary:**

This paper proposes a video-based HOI detection method that formulates the spatio-temporal representation in a video clip as tubelet tokens, in which the tubelet tokens are extracted based on the window-based multi-head self-attention with learnable offsets to aggregate semantically-related patch tokens in the spatial domain, and then temporal token linking to aggregate one-to-one matched tokens from the other frames. The tubelet tokens are then refined with global contexts and used as the video embeddings for querying HOI instances by a well-known transformer decoder, similarly to QPIC.

The tubelet tokens are better than patch tokens due to their compactness in representing spatio-temporal patterns and are more likely to align with visual dynamics from semantic instances. These two benefits are important in generating informative video embeddings for transformer-based HOI detection. Therefore, extensive experiments show that a video-based HOI detection model with tubelet tokens becomes more effective and efficient than previous patch token-based reference methods.

**Questions:**

Overall, this is a good paper with a sound novelty. I am almost satisfied with this manuscript, but it is encouraging if the authors can answer the questions listed in the paper's weaknesses.

**Limitations:**

Yes, the authors have adequately addressed the limitations and potential negative societal impact of their work.

**Strengths And Weaknesses:**

--- Strengths ---

1. Video embeddings as tokens that represent spatio-temporal tubelets are beneficial for various action-related tasks, such as HOI detection. This paper proposes a nice and practical attempt toward this goal and is a good reference for readers in related research fields.
2. This paper is generally well-written with clear organization and comprehensive experiments.
3. The effectiveness and efficiency of the proposed method have been proven, with significant performance gains and runtime speedup to existing methods.

--- Weaknesses ---

1. This paper claims that the tubelet tokens are aligned with semantic instances, but the experiments did not validate this contribution. In fact, the IWP operation is not guaranteed to extract instance-level tokens.

2. The technical details should be clarified:
- Token linking: The nms-one-hot operation seems to introduce biases since different orders of query tokens will result in the different linkage of key tokens. I am not sure whether such biases affect the prediction or not.
- Global context refining: Knowing the global contexts should be useful since contexts may help discriminate fine-grained interactions. But enhancing global contexts may not discriminate against co-occurring interactions because these interactions may be different labels describing the same spatio-temporal action patterns.

3. Experiments:
- The analysis of CNN-based & Transformer-based methods:
  - The reference methods ST-HOI and STGPN should be briefly introduced and compared with the proposed method.
  - How the experiment ``spatial distance'' is conducted is not quite sure. For example, does the spatial distance means the distance between the human and the object? How to handle the case that the spatial distance is temporally varying?
  - How to generate the similarity matrix in Fig.3 (c)? Line 252-256 briefly introduces the procedure, but it is still not quite straightforward. Moreover, why 'watch' and 'feed' are likely to co-occur?

---

> ### Author Response · Authors · 2022-08-02
> **Response to Reviewer 7mrZ [1/2]**
>
> Thanks for your valuable comments and your appreciation for our technical contributions. Below we discuss the concerns you have raised in detail.
>
> **Q1: Validation of that the tubelet tokens are aligned with semantic instances and that the IWP can extract instance-level tokens.**
>
> We have given some results for this validation in the main body of our manuscript and the supplementary material.
>
> In the supplementary material, we visualized the sampling locations of instance tokens and tubelet tokens in Figure 2 and Figure 3, respectively. Qualitatively, the envelope lines of the sampling locations are roughly aligned with semantic instances. In the main body of our manuscript, we compared HOI detection results by performing a simple decoder on instance/tubelet tokens and tokens obtained by regular window partition, respectively, in Table 2a. The large improvement brought by instance/tubelet tokens quantitatively implies that instance/tubelet tokens are roughly aligned with semantic instances. Besides, during the rebuttal period, we conduct an extra experiment to explicitly measure how precise the instance/tubelet tokens are aligned with semantic instances. We measure this by computing the IoU between each instance token and its corresponding ground truth human/object bounding box. For comparison, we also compute this IoU metric for patch tokens. To this end, we first match each patch token to a human/object instance by performing classification on patch tokens. Then we compute the IoU between each group of patch tokens matched to the same human/object instance and the ground truth human/object bounding box. We report the mean IoU over the entire testing set, where the instance tokens and the patch tokens achieve 65.30\% and 17.80\%, respectively. This comparison result shows the strong ability of instance/tubelet tokens to align with semantic instances.
>
>
> Although IWP cannot guarantee that every generated token is exactly aligned with an instance, the above qualitative and quantitative results show that IWP has a great ability to extract instance-level tokens, compared with patch tokens and tokens obtained by regular window partition. This ability comes from two key designs: 1) IWP is performed on progressively downsampled feature maps, which give a sufficiently large receptive field; 2) The stacked convolution and attentions layers have provided enough contextual information (rich features) for IWP to learn the right offsets to align with instances.
>
> ---
>
> **Q2: Does nms-one-hot introduce biases when the order of query tokens changes?**
>
> The order of query tokens does not influence the nms-one-hot operation. Since nms-one-hot operation is performed on the similarity matrix $\mathbf{A}$ (Eq. 9), we would like to give a brief description of $\mathbf{A}$ first.
>
> $\mathbf{A}$ is a three-dimensional matrix where the first dimension is index of frame, the second and third dimensions are index of query and key tokens, respectively.  Taking $t$-th frame as example, since the second dimension is the index of query token, changing the order of the query tokens is actually swapping the rows in $\mathbf{A}_t$. However, the nms-one-hot operation is looking for the maximum value along the column. Therefore, the whole process is not affected by the order of query tokens.
>
> We further verify this conclusion through an experiment. Concretely, we manually and randomly change the order of the query tokens for 3 times, and we observed no difference in the result.

---

> > ### Author Response · Authors · 2022-08-02
> > **Response to Reviewer 7mrZ [2/2]**
> >
> > **Q3: Global context refining may not discriminate against co-occurring interactions.**
> >
> > Global context refining is **NOT** used for discriminating against co-occurring interactions, but for detecting them simultaneously. Note that, HOI detection is a multi-label classification task, i.e., a human/object may have multiple interaction labels. Taking the example of  "holding-fork'' and "eating-cake'', although they describe almost the same action pattern, "holding-fork'' is prone to be detected but "eating-cake'' might be missed, if the information of "cake'' is not captured. In this case, global context refining can exchange the information between "hold-fork'' and "cake'', which leads to the label of ``eating''. Namely, with global attention refining, the detection of one interaction instances can help to detect another one. In previous literature, global context refining was shown to be helpful for detecting co-occurring interactions[20, 37].
> >
> > [20] Kim+. Hotr: End-to-end human-object interaction detection with transformers. In CVPR, 2021.
> >
> > [37] Tamura+. Qpic: Query-based pairwise human-object interaction detection with image-wide contextual information. In CVPR, 2021.
> >
> > ---
> >
> > **Q4: The reference methods ST-HOI and STIGPN should be briefly introduced**
> >
> > We have briefly introduced ST-HOI [4] in Line 58-61 and STIGPN [42] in line 55-56 in the related work and analyzed their differences with TUTOR. The comparison to these two reference methods were reported in Table 4.
> >
> > ---
> >
> > **Q5: More detailed description to the experiment "spatial distance''**
> >
> > As the datasets we used provide human/object annotations on each frame, we can compute the spatial distance, i.e., the Euclidean distance, between the center of each human and the center of the interacted object frame-by-frame.
> >
> > **Q6: How to generate the similarity matrix in Figure 3c? Why "watch'' and "feed'' are likely to co-occur?**
> >
> > We describe the detailed process of generating the similarity matrix in **G2**. We think it is easy to understand "watch'' and "feed'' are likely co-occur. Humans are normally watching the animal/baby they are feeding. As a simple validation, we count 8,000 frames with label of "feed'', and find more than 60% of them also has label of "watch''.

---

### Author Response · Authors · 2022-08-02
**General Response**

We thank all the reviewers for their valuable comments and insightful advice. Below we address several common concerns.

**G1: An instance does not always appear in all frames, how does this affect the token linking? [R2, R3]**

In practical scenarios, it is true that an instance does not always appear in all frames, especially when the video is long. To investigate the influence of this issue, we conducted the experiment in Table 1(c) and analyzed the results in Line 281-287. Specifically, when a video is short, the assumption that an instance appears in all frames does not introduce much noise, but this might not be true for long videos. As reported in Line 283, we found that when a video is longer than 16 seconds, the performance of nms-one-hot would be significantly degraded. We addressed this problem by splitting a long video into short clips with the same length and performing nms-one-hot assignment in each short clip. This strategy can alleviate this problem to some extent. Besides, in Equation 11, instead of directly concatenating the matched tokens, we summed them using the similarity $\mathbf{A}$ as the weights. In this way, if a counterfeit key token matches a query token, its similarity would be relatively small, which can reduce the effect of noise.

It is our belief that reducing redundancy in token representation is crucial for vision Transformers. Compared with patch tokens, the proposal of tubelet tokens has made considerable progress, especially for eliminating redundancy in the spatial domain. To further eliminate the redundancy in the temporal domain, more fine-gained perception of instances in each frame is required. A straightforward idea is to perform an additional object detector on each frame, and if an instance is not detected, we can use a [mask] token to indicate it. However, it will make the entire model much more complex and introduce additional computational costs. Moreover, the object detector may also have some errors. In our future work, we will keep exploring how to more effectively eliminate the redundancy of the tubelet tokens in the temporal domain.

---

**G2: A more detailed explanation for Figure 3c. [R1, R3]**

To systematically answer this question, in the following three paragraphs, we elaborate **what Figure 3c illustrates**, **how to produce Figure 3c**, and **what conclusion we can draw from Figure 3c**, respectively.

Figure 3c illustrates the attention weight map computed by the self-attention module in the last decoder layer of TUTOR. It shows that Transformer is able to model the pair-wise relations between different HOI instances, i.e., the priors of co-occurring interactions. Specifically, each decoder layer consists of a self-attention module and a cross-attention module, where the cross-attention module globally reasons about all HOI instances by using the features computed by the encoder of TUTOR as the context. In comparison, self-attention calculates pairwise attention weights between different HOI instances, which does not involve the visual features from the encoder of TUTOR.

To produce Figure 3c and make the results convincing, we picked 5 static interactions and 5 dynamic interactions, each of which has enough samples (over 10,000 frames). Then, for each input sample, we generated an attention  map $ \mathbf{A}\_{\text{self}}$ with size of $N_q \times N_q$ by the self-attention module in the last decoder layer of TUTOR, as mentioned above, where $N_q=100$ is the number of HOI queries we manually defined. Then, if two different interactions (e.g., the $i$-th and $j$-th interaction) are simultaneously predicted for an input sample, we recorded the attention weight $\mathbf{A}_{\text{self}}^{(i,j)}$. Finally, the scores shown in Figure 3c are the average attention weights over all samples.

As HOI detection is a multi-label classification task, i.e., a human/object may have multiple interaction labels, it was found that the performance of HOI detection can be improved by mining the priors of these co-occurring interactions [8,15,20]. Figure 3c evidences that Transformer can mine such priors, i.e., interactions that are more likely to co-occur have higher attention weights. This shows the superiority of Transformer in HOI detection.

---

### Author Response · Authors · 2022-08-02
**Summary of Revision**

We thank all reviews for the constructive comments. We upload a revision with a few modifications, detailed as follows:
- Double-check the manuscript and revise some typos.
- Redefine some symbols to make them more clear.
- Add an ablation study about global context refining.
- Increase the spacing between paragraphs and enlarge Figure 2 and Table 3c to make them more clear.
- Move the detailed process of calculating loss function into supplementary material.

To avoid confusion, the line numbers, equation numbers, and reference numbers used below are consistent with the reviewed version, not the re-upload revision.

We hope our response addresses your concerns and we would be glad to discuss any further questions.

---

### Meta-Review · Area_Chair_fZKK · 2022-08-29

**Recommendation:** Accept
**Confidence:** Certain

**Metareview:**

*Summary*

This paper presents a novel vision Transformer TUTOR for human-object interaction detection in videos. TUTOR structurizes a video into a few tubelet tokens by agglomerating and linking semantically-related patch tokens along spatial and temporal domains. Experiments are conducted on VidHOI and CAD-120, showing that the proposed approach is more effective and efficient than previous patch token-based reference methods.

*Reviews*

The paper received 3 reviews, with ratings: 6 (weak accept), 5 (borderline accept) and 5 (borderline accept). All reviewers voted to accept the paper, but raised some concerns:
- the claim that tubelet tokens align with semantic instances is not rigorously supported (authors added additional evidence).
- an ablation study of the global context refining mechanism is required (this has been added by the authors).
- an experimental comparison to 'Detecting Human-Object Relationships in Videos' [17/18] is required (authors note that code is not available, but instead provide a comparison with an alternative similar approach that reported even better performance).
- clarification of the relationship between this model and Deformable DETR is required
- some details of the model require clarification.

*Decision*

I am satisfied that the substantive concerns raised by reviewers have been addressed by the authors, and with all reviewers voting to accept the paper I also agree. I encourage the authors to carefully update the manuscript to address all the discussions below.


**Award:**

No

---

### Decision · Program_Chairs · 2022-09-14

Accept